# The Radial Piezoelectric Response from Three-Dimensional Electrospun PVDF Micro Wall Structure

**DOI:** 10.3390/ma13061368

**Published:** 2020-03-18

**Authors:** Guoxi Luo, Yunyun Luo, Qiankun Zhang, Shubei Wang, Lu Wang, Zhikang Li, Libo Zhao, Kwok Siong Teh, Zhuangde Jiang

**Affiliations:** 1State Key Laboratory for Manufacturing Systems Engineering, School of Mechanical Engineering, Xi’an Jiaotong University, Xi’an 710049, China; luoguoxi@mail.xjtu.edu.cn (G.L.); luoyunyun@stu.xjtu.edu.cn (Y.L.); zhangqiankun3394@stu.xjtu.edu.cn (Q.Z.); wang.lu@stu.xjtu.edu.cn (L.W.); zhikangli@mail.xjtu.edu.cn (Z.L.); zdjiang@mail.xjtu.edu.cn (Z.J.); 2Xi’an Jiaotong University Suzhou Institute, Suzhou 215123, China; wangwsb@mail.xjtu.edu.cn; 3State Key Lab of Digital Manufacturing Equipment & Technology, Huazhong University of Science and Technology, Wuhan 430074, China; 4School of Engineering, San Francisco State University, San Francisco, CA 94132, USA

**Keywords:** 3D electrospinning, PVDF fibers, piezoelectricity, energy harvesting, piezoelectric sensing

## Abstract

The ability of electrospun polyvinylidene fluoride (PVDF) fibers to produce piezoelectricity has been demonstrated for a while. Widespread applications of electrospun PVDF as an energy conversion material, however, have not materialized due to the random arrangement of fibers fabricated by traditional electrospinning. In this work, a developed 3D electrospinning technique is utilized to fabricate a PVDF micro wall made up of densely stacked fibers in a fiber-by-fiber manner. Results from X-ray diffraction (XRD) and Fourier transform infrared spectra (FTIR) demonstrate that the crystalline structure of this PVDF wall is predominant in the β phase, revealing the advanced integration capability of structural fabrication and piezoelectric poling with this 3D electrospinning. The piezoelectric response along the radial direction of these PVDF fibers is measured while the toppled micro wall, comprised of 60 fibers, is sandwich assembled with a pair of top/bottom electrodes. The measured electrical output is ca. 0.48 V and 2.7 nA. Moreover, after constant mechanical compression happening over 10,000 times, no obvious reduction in the piezoelectric response has been observed. The combined merits of high-precision 3D fabrication, in situ piezoelectric poling, and high mechanical robust make this novel structure an attractive candidate for applications in piezoelectric energy harvesting and sensing.

## 1. Introduction

Piezoelectricity has been demonstrated to be useful in energy harvesting devices and high-precision sensors due to its unique ability of mechanical-to-electrical conversion. Traditionally, piezoelectric materials can be divided into two classes, namely, ceramic and polymeric piezoelectric materials. Ceramic piezoelectric materials typically exhibit superior piezoelectric performance than that of polymeric materials due to their higher piezoelectric stress constants and coupling coefficients [1]. Piezoelectric polymers, on the other hand, have the advantages of being lightweight, low cost, biocompatible, and highly flexible, and hence find uses in specific applications that require large mechanical deformations, such as human-centered lifestyle applications, in which the rigid and brittle properties of inorganic piezoelectric ceramics cannot be easily implemented. This is the reason why the piezoelectric polymer-based microelectromechanical systems (MEMS) and microfluidics have attracted considerable attention in recent years [2]. Among the piezoelectric polymers, PVDF is undoubtedly the most application-ready material due to its high piezoelectric response [3,4]. Nevertheless, as-deposited PVDF materials usually exist in the nonpolar α crystalline phase and must be post-treated with electrical poling and mechanical stretching to obtain the polar β phase and piezoelectricity [5].

Electrospinning is a powerful and straightforward technology that utilizes high electrostatic force to produce continuous polymeric micro/nano fibers [6,7]. It has been well demonstrated that the coexisting strong electric fields and stretching force in the process of electrospinning can align the dipole moment in the PVDF fibers’ crystal, endowing the unique piezoelectricity no matter how traditional far-field electrospinning or near-field electrospinning are utilized [8,9]. Therefore, many piezoelectric energy harvesters [10,11], actuators [12], and sensors [13,14] have been developed based on electrospun PVDF fibers. Regarding far-field electrospinning, Fang et al. demonstrated the radial piezoelectricity from randomly oriented electrospun PVDF nanofibers for the first time, and developed an electrical generator based on this nanofibrous membrane [8]. Gui et al. adopted parallel electrodes and a rotating-drum collector to align Poly(vinylidene fluoride-trifluoroethylene) (PVDF-TrFE) fibers, and the axial piezoelectric response along the far-field electrospun PVDF-TrFE fibers was successfully detected [15]. Even though the piezoelectric output performance from far-field electrospun PVDF fibers (in the order of several volts and micro amperes) satisfy most modern self-powered microsystems, the low controllability over deposition of the fibers and the macroscopic scale of the functionalized film (usually above 1 cm^2^ area) limit the applications of this substitute to integrate with true micro devices. On the other hand, near-field electrospun PVDF fibers could be a good piezoelectric component in micro and nano systems concerning the high controllability and patterning capability. Chang et al. developed a PVDF nanogenerator based on one single PVDF electrospun nanofiber, which was accurately deposited on a pair of parallel electrodes using near-field electrospinning to harvest energy from mechanical vibrations along the axial direction of the fiber [9]. To amplify the electrical output performance, a near-field electrospun PVDF fibers array, with fiber spacing ca. 20 μm, was fabricated and arranged in series and in parallel to detect their axial piezoelectricity [16]. Although such a fiber array strategy could boost electrical outputs, the dispersed arrangement of fibers on the two-dimensional plane would inevitably cost more spaces and thus decrease the output power density. In the end, a method to realize high-density fiber structure, like controllable 3D arrangement, is critical to enhance piezoelectric performance for practical application.

Recently, a technique for achieving 3D near-field electrospinning on paper substrate for precise fabrication of arbitrarily shaped 3D electrospun structures has been successfully developed [17] by the authors. Using a printing paper as the fiber collector, the residual solvents from near-field electrospun fibers can infiltrate the paper substrate and a charge transfer path between the deposited fibers and ground plate can be established. Such a charge transfer grounds the locally deposited fibers and provides a low-potential site for the deposition of subsequent fibers, enabling a self-aligned fabrication process. In this work, utilizing this 3D electrospinning technology, we fabricated a free-standing, out-of-plane micro wall by densely and uniformly stacking PVDF fibers that were electrospun in a fiber-by-fiber manner on paper substrate. Then, this micro wall is toppled and laid flat on the paper, both of which are subsequently sandwiched between a pair of aluminum electrodes. The piezoelectric response along the radial direction of the fibers (normal to the wall surface) is measured. This, to the best of the authors’ knowledge, is the first time that the radial piezoelectricity from near-field electrospun PVDF fibers is demonstrated. Furthermore, compared to the existing piezoelectric devices utilizing traditional electrospun PVDF fibers, this technique promises high precision over 3D fabrication, compact integration of functionalized fibers, enabling their enhanced piezoelectric performance for direct applications into other micro devices or systems.

## 2. Materials and Methods

### 2.1. Materials

The PVDF powder (Mw = 534, 000), dimethyl sulfoxide (DMSO), acetone, and anionic surfactant Capstone FS-66 were purchased from Sigma-Aldrich (Berkeley, CA, USA) for preparation of 3D electrospinning solution. The chemicals were directly used without any further treatment. Electrically conductive aluminum tapes (3302) were purchased from 3M corporation (Berkeley, CA, USA) and used as electrodes for fabrication of devices.

### 2.2. Electrospinning Solution Preparation

Firstly, 2.1 g PVDF powder was dispersed in 3 g acetone for 30 min using a magnetic stirrer, and then 7 g DMSO and 0.5 g anionic surfactant Capstone FS-66 were added into the PVDF-acetone suspension; finally, the mixture was stirred for more than 2 h to reach a good homogeneity. The solutions were prepared and stored at room temperature and 1 atm pressure. The container with PVDF solution was sealed with Parafilm (BEMIS 01852-AB, Berkeley, CA, USA) to minimize evaporation.

### 2.3. Three-Dimensional Electrospinning for Fabrication of Micro Wall

This 3D electrospinning technique actually was built on the near field electrospinning, the spinneret-to-collector distance was set at 1 mm, allowing deposition of fibers to obtain high positional accuracy. The applied voltage was 1.2 kV. A 30 g (inner diameter 150 μm) needle was utilized as spinneret for electrospinning. A programmable x-y translational motion stage with setting speed of 100 mm/s was used to repetitively deposit fibers for out-of-plane construction, and a z-axis manual linear stage (speed ca. 7 μm/min) was set to guarantee the spinneret-to-fiber distance at a constant. With pre-designing the trajectory of this x-y translational motion stage, a 3D wall construction in the fiber-by-fiber manner could be achieved with repetitive depositions, and the height can be adjusted by altering the repetition times.

### 2.4. Characterization and Measurements

The morphology and structure of the as-prepared micro wall were observed with a field scanning electron microscope (FESEM; Gemini SEM 500, Zeiss, Heidenheim, Germany). The crystal structure was characterized by X-ray diffraction (XRD; D8 Advanced, Bruker, Karlsruhe, Germany) using Cu Kα radiation over a 2θ range of 5°–40° through reflection mode. Fourier transform infrared spectroscopy (FTIR; Nicolet iS10, Thermo Fisher Scientific, Waltham, MA, USA) was used to obtain the absorption spectra and characterize the crystallographic structure. An electrochemical workstation (Reference 600+, Gamry Instruments, Berkeley, CA, USA) was utilized for detection of output voltages and currents. The mechanically dynamic compression and decompression on the as-prepared device was provided by a mechanical shaker (MS100, YMC Piezotronics Inc., Yangzhou, China).

## 3. Results and Discussion

### 3.1. Fabricaition and Characterization

Experimentally, as illustrated in Figure 1a, PVDF fibers were electrospun into an out-of-plane micro wall structure made up of 60 densely stacked PVDF fibers, in a fiber-by-fiber manner, on paper substrate. Figure 1d shows a SEM image of the fabricated micro wall of ca. 5 μm in width, an orderly out-of-plane 3D arrangement for fibers deposition can be clearly seen. In the next step, as illustrated in Figure 1b, the micro wall structure was trimmed by dipping acetone to dissolve and eliminate some redundant parts. After the wall was toppled onto paper substrate, a consistent fiber-by-fiber manner could be observed, and the height was over 200 μm, as shown in Figure 1e. Finally, to investigate the radial piezoelectric performance, the device was assembled with aluminum tapes as a pair of top and bottom electrodes, and the electrospun PVDF wall on paper substrate as the functional layer is sandwiched in between as illustrated in Figure 1c. The paper substrate was not removed as this insulating layer can prevent an electrical short between the top and bottom electrodes. Figure 1f shows the optical image of the as-fabricated device made up of a four-layer sandwich structure of top electrode/electrospun PVDF wall/paper substrate/bottom electrode.

To demonstrate the in situ piezoelectric poling of this 3D electrospinning process, XRD and FTIR are utilized for analysis of the crystal structure, and a spin-coated PVDF is used as reference. Figure 2a shows the XRD curves; it can be seen that the spin-coated PVDF film exhibits a prominent peak at 2θ = 18.6°, indicative of the (020) plane from the non-polar α phase [18]. In contrast, the electrospun wall structure shows a relatively weak and almost negligible peak at α phase and a prominent peak at 2θ = 20.6°, which is attributed to the sum of the diffraction at (110) and (200) planes unique to β phase [19]. The β phase is further confirmed by FTIR test with clear characteristic peak at 840 cm^−1^, 1076 cm^−1^, 1280 cm^−1^, and 1431cm^−1^ as illustrated in Figure 2b. However, the spin-coated PVDF film only reflects weak peaks at 764 cm^−1^, 796 cm^−1^, 975 cm^−1^, 1209 cm^−1^, and 1385 cm^−1^, corresponding to the characteristic absorption bands of *trans-gauche* linkages, configuration of the α phase [20,21]. Both the XRD and FTIR tests demonstrate that this 3D electrospinning process can transform the nonpolar α phase into polar β phase.

### 3.2. Piezoelectric Performance

For testing the dynamic piezoelectric response, a mechanical shaker, operating at a specific frequency, was utilized to repetitively compress and decompress this device. The measurement platform is illustrated in Figure 3a, which includes a signal generator and a power amplifier to drive the shaker to work under a setting mode, and an electrochemical workstation is utilized to record the electrical outputs respectively. Figure 3b shows the optical image of this shaker. The mechanism for the piezoelectric output, normal to the electrospun wall surface, is illustrated in Figure 3c–e. Without the mechanical perturbation, no output current or voltage can be observed. As the electric dipoles randomly oscillate, the total spontaneous polarization from the electric dipoles is constant, as seen in Figure 3c. When mechanical impact is applied onto the device, each electrospun PVDF fiber is compressed, the spontaneous polarization of each fiber along the radial direction decreases significantly, a flow of electrons from the bottom electrode to the top electrode is produced due to the decrease in the induced charges, illustrated in Figure 3d. Conversely, when the impact is released, an opposite signal can be observed because the polarization is recovered, shown in Figure 3e. As such, under external mechanical perturbation, the electrospun device acts as a “charges pump” to generate electricity.

Through setting the signal generator and power amplifier, a sinusoidal mechanical compression and decompression at a frequency of 0.5 Hz and a peak force of ca. 1 N is generated to drive the device. Upon this mechanical impact, Figure 4a,b record the dynamic output voltage and current, respectively. The typical peak electrical outputs for this device are ca. 0.48 V and ca. 2.7 nA. To verify that the measured signal is really produced from piezoelectric responses instead of artificial noises, voltage and current from “switching polarity” [22] tests are measured, as shown in Figure 4c,d. When the polarity of the fabricated device connected to the measurement system is reversed, the shapes of electrical output curves are flipped, confirming the piezoelectric response. However, the values of voltage and current in the reversed connection are almost reduced by a half compared with those in the forward connection; the result is similar to the previous reports based on micro or nano piezoelectric structures [16,20,23], possibly caused by a bias current in the measurement system.

Furthermore, different amplitudes of driving force can be easily produced by adjusting the magnification factor of power amplifier, and the force sensor in the measurement system can detect the real-time value of force imposed on device. As such, the relationship between driving force and generated outputs can be investigated. When the driving force is varied from 200 to 1000 mN, the peak values of output voltage/current are ca. 0.37 V/2.1 nA, 0.41 V/2.3 nA, 0.44 V/2.45 nA, 0.46 V/2.6 nA, 0.48 V/2.7 nA, respectively, as shown in Figure 5. It can be clearly seen that the generated output is almost linear to the driving force, revealing the high potential of this 3D electrospun PVDF microstructure as piezoelectric component for force/pressure sensing.

The durability of the output performance has also been examined by constant mechanical compression and decompression upon the as-prepared device for over 3 h, which means over 10,000 mechanical cycles when the impact frequency is set as 1 Hz. The output current is recorded as shown in Figure 6, which shows that no obvious reduction in the signal is observed, revealing the high stability and robust for our 3D electrospun structure. It is worth mentioning that the current exhibits a slight increase from ca. 2.9 nA to ca. 3.1 nA after 3 h compression as shown in the bottom two panels of Figure 6, this can be attributed to the repeated mechanical compression which leads to the decrease in contact resistance between the electrospun wall and top electrode. The excellent durability can enable some specific applications, such as heel energy harvesting and tire pressure detection, in which high and uninterrupted mechanical impact exists.

The electrical response from spin-coated PVDF film on paper substrate is also measured with the sandwich-like electrodes configured under the same measurement conditions as recorded in Figure 7a,b. The output voltage and current from the spin-coated PVDF film are ca. 0.03 V and ca. 0.15 nA, respectively, both of which are more than one order of magnitude smaller than those from the 3D electrospun PVDF wall. The reason being that, without post poling, the spin-coated PVDF exists in the nonpolar α phase and no high enough piezoelectricity can be exhibited. This result further demonstrates the advantage of in situ poling of PVDF fibers through this 3D electrospinning technology, and eliminates the possibility that the measured signals are the result of the triboelectric effect [24] between aluminum electrodes and electrospun polymeric PVDF wall in this design.

### 3.3. Enhancement of Electrical Outputs

In order to increase the total electrical outputs, either series or parallel connections of the devices can be utilized to result in multiplied voltage or current outputs, respectively. The output voltages are increased to ca. 1.35 and ca. 2.1 V when three and five devices are connected in series, respectively, as illustrated in Figure 8a. Similarly, Figure 8b shows the enhanced outputs of current by making parallel connections of devices when three and five devices are connected in parallel, the measured output currents constructively add and rise to ca. 7.1 and ca. 10.9 nA, respectively. As such, the net electrical outputs are almost linearly correlated to the number of piezoelectric components through “linear superposition” [22], the other criterion, which verifies that the electrical outputs are indeed produced by the piezoelectric effect.

## 4. Conclusions

In summary, an advanced process combining high-precision 3D fabrication and in situ piezoelectric poling in a single step is demonstrated using the 3D electrospinning technology. The out-of-plane electrospun PVDF fibers are consistently stacked in a fiber-by-fiber manner on paper substrate, while the polar β phase is produced due to the electrical poling and the mechanical stretching during electrospinning. Piezoelectric responses along radial direction of the fibers are subsequently measured. Both “switching-polarity” and “linear superposition” criteria are tested to demonstrate that the measured signals are really generated due to piezoelectric response rather than from the noises in the measurement system. The combined merits of high controllability over fabrication, in situ poling of piezoelectricity, high integration of functionalized fibers, and excellent durability of output performance make this 3D electrospun structure promise great potential to integrate with other processes and structures as energy harvesting or high-precision sensing component.

## Figures and Tables

**Figure 1 materials-13-01368-f001:**
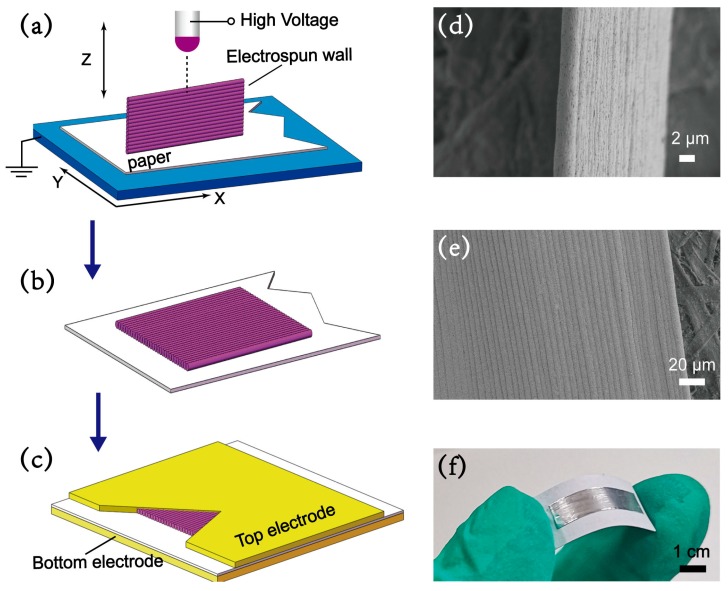
(**a**) Schematic of the fabrication process of wall structure by 3D electrospinning. (**b**) The wall is manually toppled and laid flat onto to paper substrate. (**c**) The toppled wall and paper are sandwiched in between a pair of top and bottom aluminum electrodes. SEM images of (**d**) straight wall and (**e**) toppled wall on paper substrate. (**f**) optical image of the assemble device.

**Figure 2 materials-13-01368-f002:**
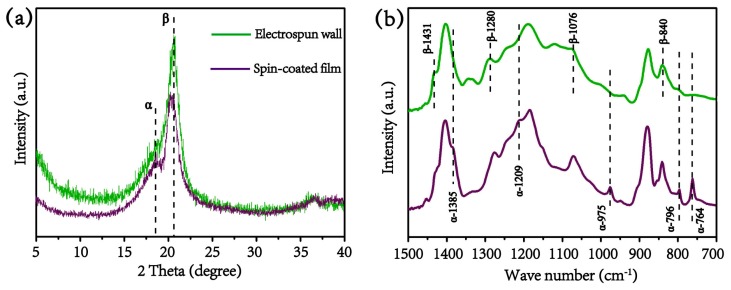
(**a**) X-ray diffraction XRD patterns and (**b**) Fourier transform infrared (FTIR) spectra for comparison between electrospun PVDF wall structure and spin-coated PVDF film.

**Figure 3 materials-13-01368-f003:**
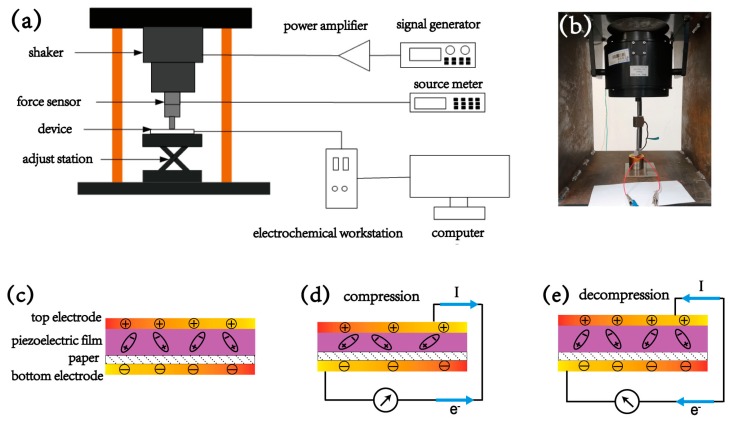
(**a**) Schematic of measurement platform for generation of dynamic compression and detection of electrical outputs. (**b**) Optical image of the mechanical shaker. (**c**) Schematic of the electric dipoles formed in the electrospun PVDF micro wall film. Principles of piezoelectric output under the (**d**) compression and (**e**) decompression mode.

**Figure 4 materials-13-01368-f004:**
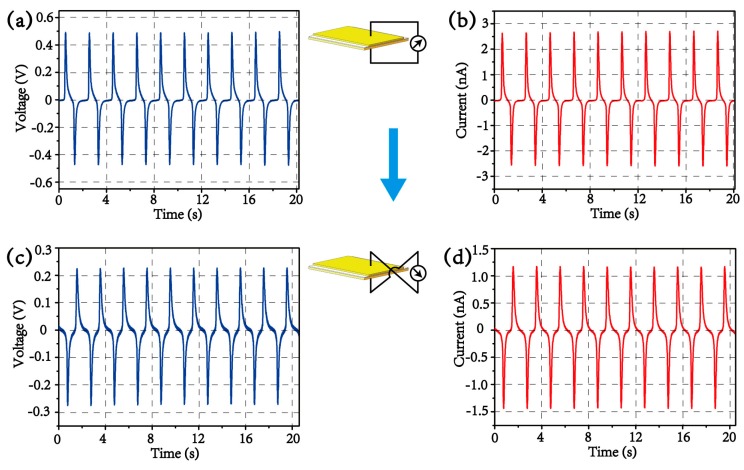
The electrical output of (**a**) voltage and (**b**) current under cyclic compression at 0.5 Hz. The switching polarity tests for output of (**c**) voltage and (**d**) current in the reversed connection. The inset is the schematic for the “switching polarity” tests.

**Figure 5 materials-13-01368-f005:**
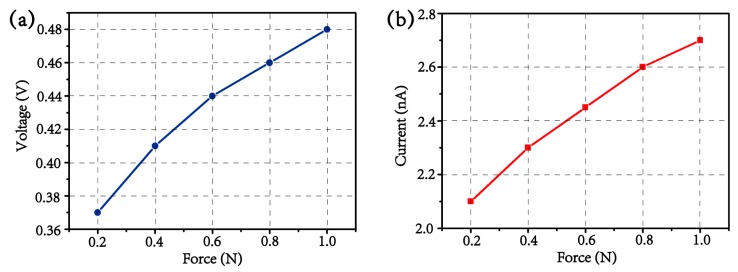
The curves of output (**a**) voltage and (**b**) current under different driving forces.

**Figure 6 materials-13-01368-f006:**
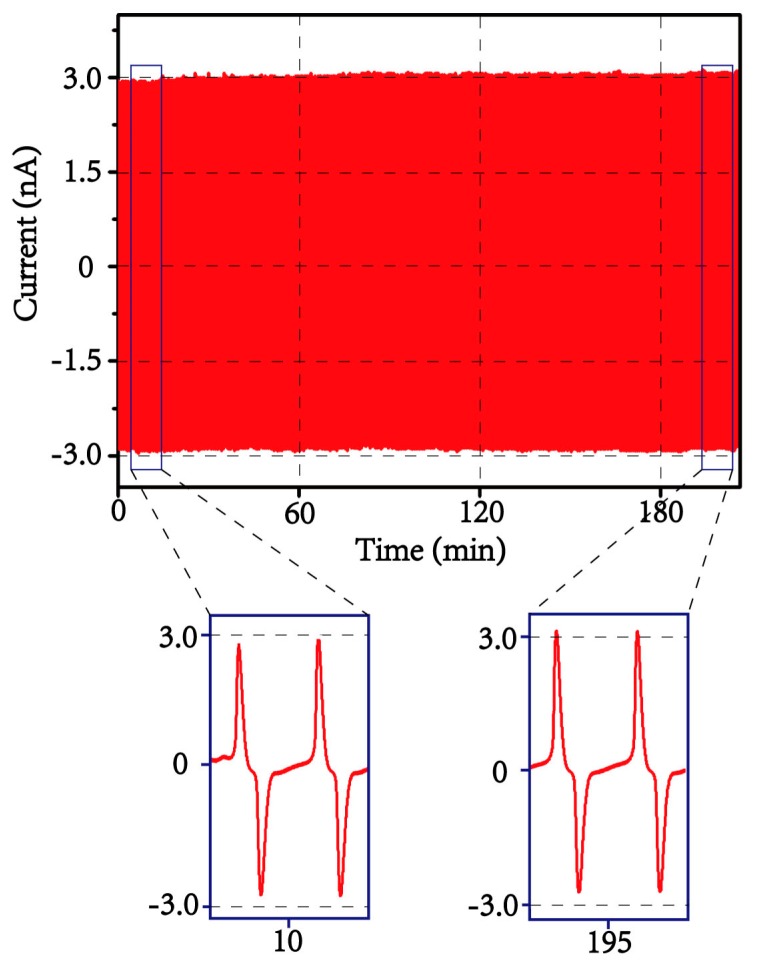
The current output curve of the as-prepared device operated for over 3 h under 1 Hz mechanical compression, the bottom two panels show the detailed shapes of current outputs.

**Figure 7 materials-13-01368-f007:**
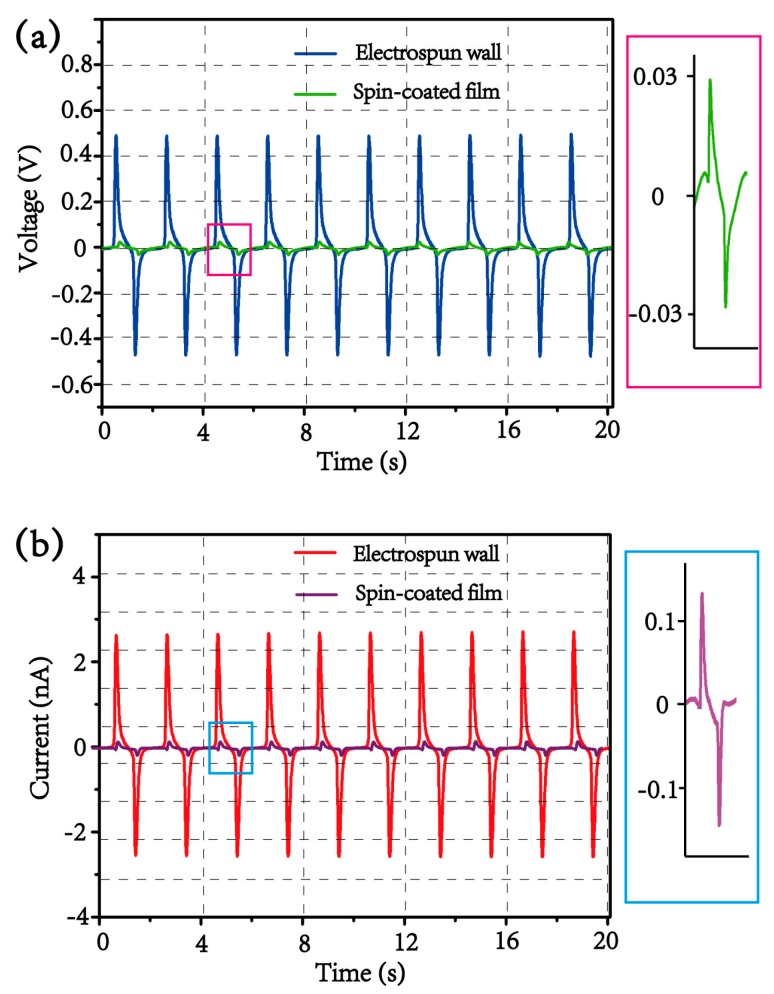
The outputs of (**a**) voltage and (**b**) current comparison between the electrospun PVDF wall and spin-coated PVDF film.

**Figure 8 materials-13-01368-f008:**
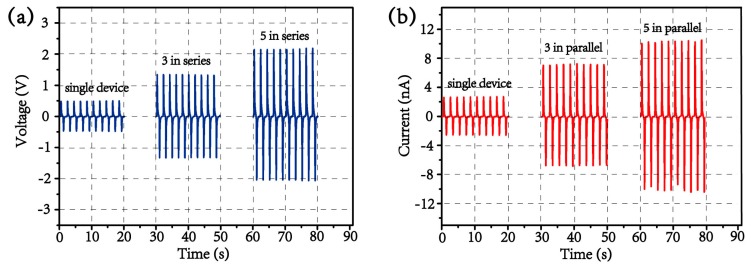
(**a**) Output voltages for one, three, and five devices that are electrically connected in series, and (**b**) output currents for one, three, and five devices that are electrically connected in parallel.

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
