# Peer review of "The Radial Piezoelectric Response from Three-Dimensional Electrospun PVDF Micro Wall Structure"

_materials, 2020, doi:10.3390/ma13061368_

Round 1

Reviewer 1 Report

A 3D electrospun  PVDF micro wall structure by applying near-field electrospinning (NFES) was previously reported by the same author in ACS Appl. Mater. Interfaces 2015, 7, 27765-27770. Even though the paper was listed in the references, it was cited for the confirmation of (020) plane from the non-polar α phase. Indeed, it should clearly be mentioned that the fabrication of 3D electrospun PVDF fibers was previously reported.

"Widespread applications of electrospun PVDF as an energy conversion material, however, has not materialized due to the random arrangement of
 fibers fabricated by traditional electrospinning. In this work, PVDF fibers are electrospun into a three dimensional (3D) micro wall structure made up of densely stacked fibers, in a fiber-by-fiber manner, on paper substrate." The concept was previously reported!

Reviewer 2 Report

I have no comments on the submitted manuscript and therefore I agree with the publication in this form.

Reviewer 3 Report

This paper introduces the fabrication of 3D PVDF fibers as piezoelectric sensing material. Overall, the work is organized and neat. However, it needs minor modifications before publishing as follows:

1- A real photo of spinning setup "not a schematic one" should be presented with zooming in the tiny space between both feeder and collector.

2- A critical comparison between the generated fibers and the famous 2D PVDF nanofibers is missing. For instance, is it possible to try one layer with 20 kV, with 10 cm distance and compare the generated 2D nanofibers thin mat with the 3D microfibers?

3- Is it possible to add more analysis regarding the piezoelectric characteristics, such as relation between force/pressure and the generated voltage or the piezoelectric coefficient analysis d33?

4- Why did not the authors try other process parameters to generate the optimum sample with highest possible voltage/current amplitudes, such as applied voltage, distance, pumping rate....etc?

Reviewer 4 Report

Comments to materials-734114

Please prepare a revision by reflecting the following comments faithfully.

  1. I recommend you to remove ‘radial’ in the title, because the piezoelectric response from even far-field electron spun PVDF nanofiber webs is also generated by radial directional deformation, i.e., thickness change while considering individual nano fiber under the cyclic pressure normal to the surface the nanoweb. Radial piezoelectric response is not unique to your sample.
  2. In many other papers regarding near-field electrospinning of PVDF, they controlled flow rate to get a good near-field electrospun fiber. However, you did not mention how to control the flow rate of the PVDF solution. How did you control it? Describe it in detail.
  3. You did not mention how to get FT-IR spectra. Which method did you use, transmission mode, reflection mode, or ATR mode? Write the method in detail in the text.
  4. You did not mention how to get XRD curves. Which method did you use, transmission mode or reflection mode? Write the method in detail in the text.
  5. Line 121-123 : You wrote “Fourier transform infrared spectroscopy is used to characterize the dipole orientation and crystallographic structure” in the text. From your IR spectra, you cannot characterize dipole orientation, but only crystal phase type. So you must delete “the dipole orientation”.
  6. Line 130 : You wrote “Figure 1d shows SEM image of the fabricated micro wall of ca. 5 μm in width”. The width appears at least 16 μm based on the scale bar in Figure 1d. Correct it.
  7. When we make a piezoelectric sensor using a poled PVDF film, we never insert any insulating layer between the PVDF film and the electrode in order to maximize the output electric signal from the PVDF film. Why didn’t you remove the paper substrate prior to making a piezoelectric sensor with near-field electrospun PVDF 2D-wall? Is there any special reason for that? If so, describe why in the text.
  8. Line 159-161 : The experimental results obtained from the general XRD and FTIR techniques used in this study could not give any evidence for piezoelectric poling during electrospinning, but only information about the preferential formation of polar β-phase rather than α-phase. The experimental poling evidence can be obtained only from the confirmation of the preferential CF2 dipole orientation toward the external electric field using a specific FTIR measurement (Refer to Polymer, 46, 12410 (2005) and Polymer, 51, 6319 (2010)),. Therefore, correct the statement in Line 161 like “highlighting the strength of 3D electrospinning as a single-step process capable of high-precision 3D fabrication”.
  9. In order to get a piezoelectric response from your PVDF samples, they must have a remnant polarization. There is no doubt about the presence of remnant polarization from the generated piezoelectric output from your sample. As proven in reference 10 you cited, the remnant polarization direction is normal to the surface of the far-field electrospun P(VDF-TrFE) nanofiber web. Therefore the polarization direction is along z-direction in Figure 1(a). But, once the wall is toppled and laid flat onto paper substrate, the polarization direction of the toppled wall should be along the x-y plane. In that case, we do not know which direction should show positive charge, upper surface or lower surface. Clarify this based on the experimental results.
  10. There are so many ways to impart cyclic force to the samples. How did you impart cyclic force to the samples? Which mode did you use, square pulse wave mode, sinusoidal wave mode, or triangular wave mode. Write the method in detail in the text.

Round 2

Reviewer 1 Report

The revised manuscript can be accepted in the present form.

Author Response

Thank you for your approval of our manuscript.

Reviewer 4 Report

Since a revision has been prepared after considering the reviewer's comments, it can be accepted in current form.